# Ultrasound Evaluation of the Rectus Femoris for Sarcopenia in Patients with Early Subacute Stroke

**DOI:** 10.3390/jcm10143010

**Published:** 2021-07-06

**Authors:** Yongmin Choi, Sun Im, Geun-Young Park

**Affiliations:** 1Department of Rehabilitation Medicine, Dongsan Hospital, School of Medicine, Keimyung University, Daegu 42601, Korea; ymchoi@dsmc.or.kr; 2Department of Rehabilitation Medicine, Bucheon St. Mary’s Hospital, College of Medicine, The Catholic University of Korea, Seoul 14647, Korea; lafoliamd@gmail.com

**Keywords:** stoke, sarcopenia, rectus femoris, appendicular lean muscle, echo intensity, ultrasound, dual-energy X-ray absorptiometry

## Abstract

We investigated the ultrasound characteristics of the rectus femoris for sarcopenia detected by dual-energy X-ray absorptiometry (DEXA) in the early subacute stroke phase. Physical features (age, sex, body mass index, and circumference of thigh) and performances (modified Barthel index in Korean, functional ambulation categories, and mini-mental state examination in Korean) were measured. The thickness of the fat layer, the thickness of the rectus femoris (TRF), echo intensity (EI), EI to TRF ratio, and strain ratio of elastography (SRE) were measured by ultrasound in 30 patients with first-ever stroke (male: *n* = 20). Appendicular lean body mass was measured by DEXA. Sarcopenia was defined according to the Foundation for the National Institutes of Health Sarcopenia Project. In total, 14 patients were in the sarcopenia group, and 16 were in the non-sarcopenia group. Clinical characteristics were similar between the two groups. In the sarcopenia group, TRF was significantly decreased in the paretic (*p* < 0.026) and non-paretic sides (*p* < 0.01), and the EI to TRF ratio on the paretic side was significantly increased (*p* < 0.049). Multivariate binary logistic regression showed that TRF on the non-paretic side was independently and significantly associated with sarcopenia (OR = 0.616, 95% CI: 0.381–0.996). The EI and SRE were not significant between the two groups. In the early subacute stroke phase, TRF on the non-paretic side is a key factor for quantitative evaluation of sarcopenia, and the EI to TRF ratio on the paretic side is also a meaningful qualitative evaluation of sarcopenia.

## 1. Introduction

Stroke survivors not only have to contend with normal aging, but also several deteriorating factors such as denervation, disuse, inflammation, dysphagia, muscle remodeling, and spasticity [1,2,3]. Skeletal muscle loss (sarcopenia) and fat infiltration into the muscle (myosteatosis) occur early and rapidly, within 1–3 weeks after stroke [4,5,6]. Six months later, sarcopenia and myosteatosis occur on the hemiplegic and non-hemiplegic sides [7,8]. Therefore, early detection of the change in muscle quality (i.e., severity of sarcopenia and myosteatosis) is important because these changes weaken muscle power and hinder functional recovery, although the main effector to overcome disability [9,10,11].

The hand grip test, walking velocity, and the sit-to-stand test can be used to evaluate sarcopenia [12,13]. However, most of these methods are difficult to execute and evaluate in stroke survivors because of hemiplegia [14,15,16]. Some imaging studies have used dual-energy X-ray absorptiometry (DEXA), magnetic resonance imaging (MRI), computed tomography (CT), and ultrasound to evaluate sarcopenia. Among them, ultrasound has some advantages, such as low cost, ease of use, and being the only available modality for use at the bedside [17,18].

Ultrasound can analyze muscle quantitatively and qualitatively by measuring the muscle thickness to assess mass, the echo intensity (EI) to assess density, which is dependent on fat infiltration, the EI to the thickness of the rectus femoris (TRF) ratio to assess the severity of intramuscular fat infiltration, and strain ratio of elastography (SRE) to assess muscle stiffness [8,19,20]. The rectus femoris can sensitively reflect the loss of lean mass, and the TRF correlates well with DEXA [5,21]. Recently, the Foundation for the National Institutes of Health (FNIH) Sarcopenia Project has proposed new definitions to identify cutoff points for appendicular lean mass (ALM) (<19.75 kg in men and <15.02 kg in women) by DEXA, which can discriminate the presence or absence of muscle weakness [22].

We assumed that the lower the ALM, the worse the degeneration in the rectus femoris would become. An ultrasound study would be able to detect degenerative changes in the rectus femoris in the early subacute stroke period. TRF would decrease according to the loss of ALM. EI would increase according to fatty infiltration into the muscle. Therefore, the EI to TRF ratio would clearly increase according to the severity of myosteatosis. SRE would change with muscle degeneration. However, there are few reports on these factors in the early subacute stroke period. Thus, we evaluated the ultrasound characteristics of the rectus femoris for sarcopenia measured by DEXA and compared it with patients without sarcopenia in the early subacute stroke phase.

## 2. Materials and Methods

### 2.1. Subjects

We performed a cross-sectional study of patients who had been transferred either from the stroke unit or the neurosurgical unit to the Department of Rehabilitation Medicine between May 2016 and August 2017. Data acquisition and analysis were reviewed and approved by the institutional review board (HC16OISI0028) and performed in accordance with the recommendations of the Helsinki Declaration of 2013 and registered with Clinicaltrials.gov (NCT02793362).

We recruited 30 first-ever stroke patients (20 male, 10 female) aged over 40 years who were admitted to a university hospital, whose lesions were confirmed by brain imaging (computed tomography (CT) or MRI), and who were transferred to our department following conventional rehabilitation protocols.

The inclusion criteria were as follows: patients with subacute stroke for the first time whose lesions were confirmed using brain imaging (MRI/CT); patients who had unilateral hemiplegia; patients who could follow 1 step of the mini-mental state examination in Korean (MMSE-K) ≥ 10; patients who did not have difficulty in premorbid ambulation or activities of daily living without devices; patients who could eat orally; and patients with functional ambulation category scores of at least 2.

The exclusion criteria were as follows: patients with double hemiplegia; patients with amputated lower limbs; patients with a fracture or surgical history in lower limbs that led to gait disturbance; patients with any surgeries that led to malnutrition; and patients with other systemic diseases that could affect the loss of muscle mass or malnutrition (hypogonadism, hypercortisolism, hyperthyroidism, growth hormone deficiency, chronic obstruction pulmonary disease, chronic kidney disease, liver cirrhosis, chronic heart failure, neurodegenerative diseases, inflammatory diseases, acquired immune deficiency syndrome, and malignancy). Additionally, patients had no spasticity according to the modified Ashworth scale or contracture in both lower limbs.

### 2.2. Measurements

All patients included in this study underwent DEXA and ultrasound performed within 1 week after transfer to our department, before beginning rehabilitation therapy. The measurement time was recorded as post onset. The interval between DEXA and ultrasound did not exceed 24 h, and patients were allowed to have sufficient rest (>15 min) before the test. The circumference of the thigh was measured at a level between half of the anterior superior iliac spine to the apex of patella on both paretic and non-paretic sides. Occupational and physical therapists who were blinded to the patients’ details measured the modified Barthel index in Korean, MMSE-K, and functional ambulation categories.

### 2.3. Assessment of Lean Mass and Definition of Sarcopenia

Body composition was measured by DEXA using a Lunar Prodigy Advance (GE Healthcare, Chicago, IL, USA). Total and segmental fat-free lean mass and fat masses were recorded. Sarcopenia was defined using the criteria of the FNIH: <19.75 kg in men and <15.02 kg in women [22].

### 2.4. Ultrasound Image Recording and Estimation

Ultrasound was performed by one sonographer who did musculoskeletal ultrasound for three years. He was blinded to the patients’ medical information. All patients laid supine with legs completely relaxed. Cross-sectional ultrasound images were recorded on half of the distance from the anterior superior iliac spine to the apex of the patella, using a HI VISION Avius (Hitachi, Tokyo, Japan) with an 18–5 MHz linear array transducer (EUP-L75). To obtain a uniform B-mode ultrasound image, the compensation function was turned off and the B-gain was set at 15 dB, the B-dynamic range at 75 dB, the depth at 5 cm, and the depth was adjusted only if the femur was not visible to confirm the maximum intensity of the femur cortex [23]. A gel pad applicator was used to avoid excessive compression of the skin and subcutaneous layer, and to obtain a more uniform strain force for elastography. The transducer was positioned perpendicular to the transverse scan for EI and to the longitudinal scan for SRE [23,24].

Three consecutive static images were acquired and averaged to minimize measurement variability. Maximal muscle thickness was measured and averaged in transverse scans as the distance between the superficial and deep fascia by using the electrical caliper provided by the device. A region of interest without surrounding fascia was chosen and calculated (black = 0, white = 255) by using a computer-assisted grey-scale analysis offered by ImageJ 1.50 g (National Institutes of Health, Bethesda, MD, USA) for EI. EI to TRF ratio was also calculated to assess the severity of myosteatosis [25,26].

Elastography of the rectus femoris was performed using a minimal local strain force by a handheld method, 3–4 Hz by piston motion, and strain force was confirmed by the visual indicator provided by the device; a uniform pattern was maintained for each patient. For real-time elastography, we selected three random static optimal images by the device. SRE was calculated (blue = 0, red = 255), and stiffness of the rectus femoris for the subcutaneous tissue (elasticity of subcutaneous tissue/elasticity of rectus femoris) was obtained and averaged by using integrated machine software (Hitachi, Tokyo, Japan, version 1.05.03.0626) (Figure 1) [27].

### 2.5. Statistical Analyses

Continuous and categorical variables were expressed as the mean ± SD and *n* (%), respectively. Statistical analyses were performed using SPSS Statistices for Windows, version 26 (IBM Corp., Armonk, NY, USA), and the level of statistical significance was defined as *p* < 0.05.

A sample size analysis for unpaired *t*-test and linear multiple regression test was performed using the statistical package G-power 3.1™ and with an effect size set at = 1.2 for the two-tailed unpaired *t*-test and to 0.5 for linear multiple regression with 4 predictors and *p* = 0.05 [28]. This revealed that the study would require at least 12 subjects per group for the unpaired *t*-test and a total sample size of at least 30 for the multiple linear regression analysis to reach a power value of 0.8, in reference to previous related studies on TRF, EI, and the EI to TRF ratio [19,21,26,29]. All variables demonstrated normal data and variance distributions based on the Kolmogorov–Smirnov test and Levene’s tests, respectively. Mean differences between the sarcopenia and the non-sarcopenia groups were analyzed with unpaired *t*-tests. Binary logistic regression analysis was used to determine the factors associated with sarcopenia in the early subacute stroke patients. Factors that returned *p* < 0.15 in univariate analyses were included in the multivariate logistic regression analysis. The odds ratio (OR) was considered to be significant if the 95% confidence interval (95% CI) did not include 1.0.

## 3. Results

Of 30 subjects enrolled in this study, 14 were categorized into the sarcopenia group and the rest into the non-sarcopenia group. The demographics and clinical characteristics of the subjects are listed in Table 1.

None of the clinical characteristics were significant between the two groups. DEXA measurements are summarized in Table 2. Lean mass was significantly decreased (*p* < 0.001) in the sarcopenia group, but fat mass was not. The main loss of lean mass occurred in both legs.

In the ultrasound measurements, the TRF in the paretic (*p* < 0.026) and non-paretic sides (*p* < 0.01) decreased in the sarcopenia group. The thickness of the fat layer and EI were not significant between the two groups. However, EI had a tendency to increase depending on muscle degeneration. Therefore, only the EI to TRF ratio on the affected side in sarcopenia (*p* < 0.049) was significant between the two groups. SRE in the sarcopenia group increased slightly compared with that in the non-sarcopenia group, but this was not significant (Table 3).

To identify possible risk factors for sarcopenia, we performed binary logistic regression analysis according to the cutoff points of sarcopenia from the FNIH Sarcopenia Project, i.e., <19.75 kg in men and <15.02 kg in women (Table 4) [22]. Multivariate logistic analysis showed that TRF on the non-paretic side was an independent and significant risk factor for sarcopenia (OR = 0.616, 95% CI: 0.381–0.996, *p* = 0.048) in the early subacute stroke phase.

## 4. Discussion

In this study, there were no significant differences in the clinical (age, sex, BMI, onset period, and circumference of the thigh) and functional characteristics (modified Barthel index in Korean, functional ambulation categories, and MMSE-K) between the sarcopenia and the non-sarcopenia groups in the early subacute stroke phase. DEXA showed that the loss of lean mass occurred in both the upper and lower extremities, but predominantly in the lower extremities. Ultrasound showed that TRF in the sarcopenia group significantly decreased in both the paretic and non-paretic sides. EI was not significant between the two groups but showed a tendency to increase according to muscle degeneration. As EI increased and TRF decreased, the EI to TRF ratios clearly increased in both groups. However, only the EI to TRF ratio in the paretic side in the sarcopenia group was significant. SRE also tended to increase with muscle degeneration, but unlike the EI to TRF ratio in the paretic side in the sarcopenia group, there was no significant change between the two groups.

Aging-related muscle loss occurs site specifically, particularly in the quadriceps and abdominal muscles, and the prevalence of sarcopenia is related to loss of the proximal leg muscle [30,31]. In critical illnesses, such as stroke, the protein breakdown in the legs is increased, and the cross-sectional area of the rectus femoris is decreased due to decreased protein synthesis [5]. In healthy older adults, protein synthesis decreases, the lean mass of the leg is reduced, and the muscle power is weakened after only 10 days of bed rest [32]. Various changes such as inactivity or immobilization by stroke, systemic inflammatory activation, accumulation of reactive oxygen species, and motor unit denervation due to brain injury occurred, and these changes accelerate sarcopenia [2,33]. Several previous studies have reported that the lean mass of the hemiparetic leg is decreased more than that of the non-hemiparetic leg in the chronic stroke phase [26,34,35]. Although there have been few studies of the subacute stroke period (within 4 weeks), the present study showed a similar result in the early subacute stroke period. We also found that the non-paretic TRF was suitable for estimating ALM associated with sarcopenia.

Myosteatosis begins within 1–3 weeks after stroke, and EI is related to fat or connective tissue within the muscle [4,5,6]. In a previous study on the chronic stroke phase, the hemiplegic and non-hemiplegic mid-thigh muscle was decreased and intramuscular fat was increased. As the muscle change worsened, TRF decreased and fat infiltration increased [21]. Another study of the correlation of EI and muscle biopsy performed in neuromuscular disease revealed that myosteatosis is the main cause of increased EI and that intramuscular fibrosis did not significantly affect EI [36]. A recent study reported that it is similar to EI and intermuscular adipose tissue measured by CT [37]. In CT, the ratio of the low-density lean tissue to muscle area revealed a greater amount of intramuscular fat relative to muscle area and the ratio was higher in the hemiplegic leg depending on the severity of myosteatosis [26]. On an ultrasound, low density reflects EI and muscle area reflects TRF. Thus, the EI to TRF ratio in ultrasound indicates the severity of myosteatosis. In this study, EI tended to increase with muscle degeneration, but there was no significant difference between the two groups. Among measurements of the EI to TRF ratio, only the EI to TRF ratio in the paretic side in the sarcopenia group was significant because EI increased and TRF decreased with muscle degenerative changes. Changes in EI and TRF moved in opposite directions according to the degenerative changes in the muscle. Therefore, even when intramuscular fat deposition was not clearly observed, the EI to TRF ratio was sensitively reflected according to the degenerative changes of the muscle in the early subacute stroke phase. The EI to TRF ratio on the paretic side could be a meaningful qualitative evaluation of sarcopenia.

SRE increased as the EI to TRF ratio increased, but there was no significant differences. Neither the fat mass of DEXA nor the thickness of the fat layer of the ultrasound was significant in the sarcopenia group. Although the EI to TRF ratio was increased by muscle degenerative change, SRE was not significant because it depended on the change in stiffness of the rectus femoris in the present study. After stroke, paralysis interferes with daily activities and causes patients to become immobilized. In an animal model, the immobilized limb begins to show muscle atrophy within hours, and increased perimysial connective tissue and fat [38,39]. This also occurs in humans [40]. Intramuscular fat infiltration increases the stiffness of the muscle [41]. However, it was too early to observe changes in muscle stiffness related to intramuscular fat deposition in this study because only the tendency of increase in EI was evaluated.

The limitations of this study included the sample size, which was appropriate for linear multiple regression analysis but was insufficient to estimate cutoff values according to each sex. Further research would be needed to identify the cutoff values for each sex. The possibility of sarcopenia before stroke could not be completely ruled out. We attempted to minimize the possibility of sarcopenia by using strict exclusion criteria and by investigating the pre-stroke activity status of patients and their families. We did not perform muscle biopsy, MRI, or CT to estimate the severity of myosteatosis in this study. However, previous studies have reported on the correlation between myosteatosis and EI [21,35]. According to a recent report, EI and fat infiltration within muscle measured by CT were similar to those measured by ultrasound [36].

## 5. Conclusions

In the early subacute stroke phase, TRF in the non-paretic side is a key factor for the quantitative evaluation of sarcopenia. The EI to TRF ratio in the paretic side is a meaningful qualitative evaluation of sarcopenia. Therefore, ultrasound evaluation of the rectus femoris muscles on both paretic and non-paretic sides would be useful to monitor for sarcopenia at the bedside.

## Figures and Tables

**Figure 1 jcm-10-03010-f001:**
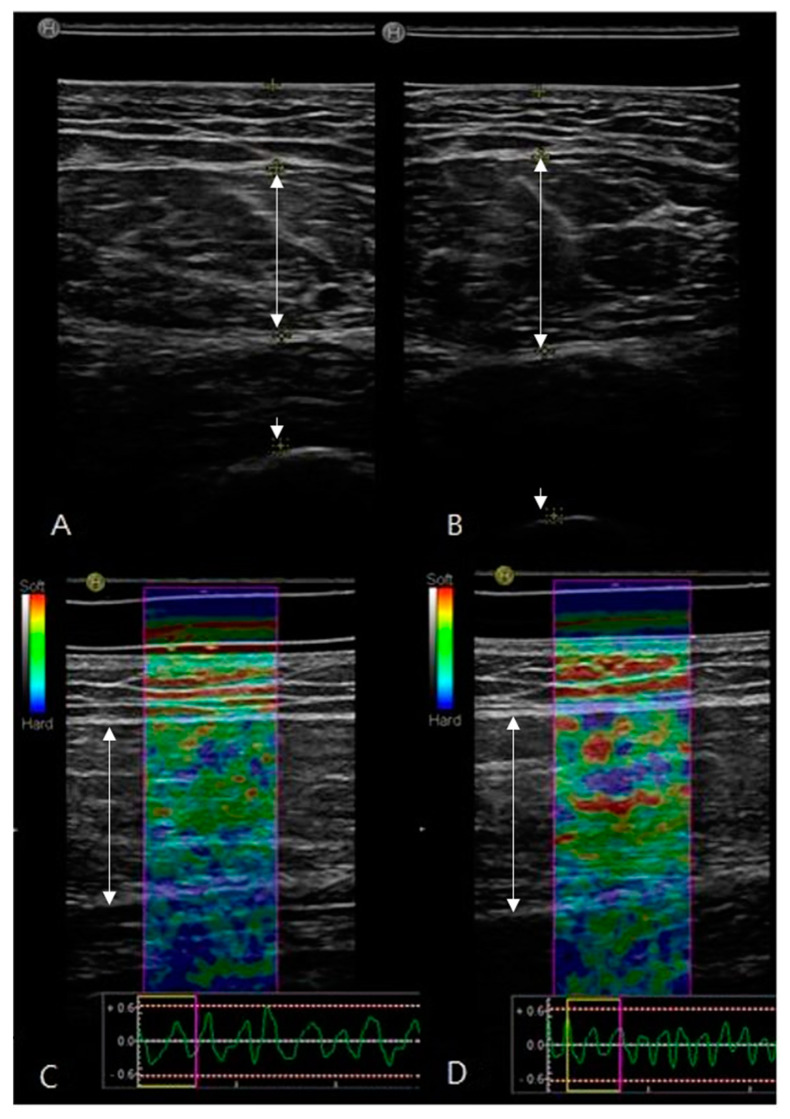
This is an example of an ultrasound of rectus femoris. The single arrow is the cortex of the femur and the double arrow is the rectus femoris. (**A**,**C**) B-mode transverse scan for the thickness of the rectus femoris (paretic 14.51 ± 2.47, non-paretic 15.31 ± 2.49), echo intensity measurements (paretic 43.94 ± 15.58, non-paretic 40.35 ± 14.45), and real-time strain elastography longitudinal scan with a pressure wave for strain ratio of elastography (paretic 1.86 ± 1.06, non-paretic 1.68 ± 0.77) showed in the sarcopenia group. (**B**,**D**) B-mode for the thickness of the rectus femoris (paretic 17.32 ± 3.83, non-paretic 18.50 ± 3.66), echo intensity measurements (paretic 35.75 ± 11.59, non-paretic 34.51 ± 15.52), and real-time strain elastography with a pressure wave for strain ratio of elastography (paretic 1.41 ± 0.63, non-paretic 1.36 ± 0.53) showed in the non-sarcopenia group.

**Table 1 jcm-10-03010-t001:** Clinical characteristics between subjects with and without sarcopenia.

	Sarcopenia (*n* = 14)	Non-Sarcopenia (*n* = 16)	*p*-Value
Age (year)	69.79 ± 9.82	65.38 ± 12.29	0.291
Male	9 (64.3%)	11 (68.8%)	0.804
Post onset (days)	19.77 ± 7.07	26.85 ± 13.35	0.108
Body Mass Index (kg/m^2^)	22.52 ± 3.23	24.38 ± 3.02	0.126
pCircumference of Thigh (cm)	36 ± 2.12	36.5 ± 3.32	0.643
npCircumference of Thigh (cm)	37 ± 5.57	36.63 ± 3.48	0.826
Modified Bathel index in Korean	51.21 ± 25.03	60.19 ± 27.35	0.359
Functional ambulation categories	2.57 ± 1.09	2.81 ± 2.81	0.499
MMSE-K	22.69 ± 5.06	22.81 ± 6.48	0.957

Values are presented as mean ± SD or *n* (%); p, paretic; np, non-paretic; MMSE-K, mini-mental state examination–Korean.

**Table 2 jcm-10-03010-t002:** DEXA measurements between subjects with and without sarcopenia.

	Sarcopenia (*n* = 14)	Non-Sarcopenia (*n* = 16)	*p*-Value
pLean Mass of Arm (kg)	2.08 ± 0.39	2.75 ± 0.41	0.000
npLean Mass of Arm (kg)	2.15 ± 0.41	2.92 ± 0.48	0.000
pLean Mass of Leg (kg)	5.70 ± 1.01	7.65 ± 1.36	0.000
npLean Mass of Leg (kg)	5.85 ± 0.90	7.85 ± 1.43	0.000
Appendicular Lean Mass (kg)	15.81 ± 2.68	20.86 ± 3.70	0.000
pFat Mass of Arm (kg)	0.72 ± 0.46	0.77 ± 0.48	0.762
npFat Mass of Arm (kg)	0.74 ± 0.47	0.81 ± 0.47	0.709
pFat Mass of Leg (kg)	2.21 ± 1.04	2.17 ± 1.26	0.933
npFat Mass of Leg (kg)	2.25 ± 1.12	2.22 ± 1.22	0.942
Total Fat Mass (kg)	16.56 ± 8.16	17.59 ± 8.18	0.742

Values are presented as mean ± SD; p, paretic; np, non-paretic.

**Table 3 jcm-10-03010-t003:** Ultrasound measurements between subjects with and without sarcopenia.

	Sarcopenia (*n* = 14)	Non-Sarcopenia (*n* = 16)	*p*-Value
pThickness of Fat Layer (cm)	10.44 ± 5.65	8.39 ± 4.23	0.304
npThickness of Fat Layer (cm)	9.98 ± 5.27	8.60 ± 4.11	0.465
pTRF (cm)	14.51 ± 2.47	17.32 ± 3.83	0.026
npTRF (cm)	15.31 ± 2.49	18.50 ± 3.66	0.010
pEI	43.94 ± 15.58	35.75 ± 11.59	0.111
npEI	40.35 ± 14.45	34.51 ± 15.52	0.297
pSRE	1.86 ± 1.06	1.41 ± 0.63	0.166
npSRE	1.68 ± 0.77	1.36 ± 0.53	0.200
pEI to TRF ratio	3.21 ± 1.58	2.23 ± 1.00	0.049
npEI to TRF ratio	2.81 ± 1.40	2.03 ± 1.27	0.124

Values are presented as mean ± SD; p, paretic; np, non-paretic; EI, echo intensity; TRF, thickness of rectus femoris; SRE, strain ratio of elastography.

**Table 4 jcm-10-03010-t004:** Risk factor of US measurements according to sarcopenia.

	Univariate Analysis					Multivariate Analysis				
		B	OR	*p*-Value	95% CI		B	OR	*p*-Value	95% CI
Sarcopenia M < 19.75 F < 15.02 (kg)	Age	0.37	1.038	0.283	0.970–1.111	Age	−0.60	0.971	0.316	0.837–1.392
	Male	−0.201	0.818	0.796	0.179–3.744	Male	0.664	1.943	0.542	0.230–16.394
	Body Mass Index	−0.208	0.812	0.135	0.618–1.067	Body Mass Index	−0.29	0.971	0.873	0.667–1.392
	pThickness of Fat Layer	0.09	1.094	0.298	0.924–1.296	npTRF	−0.485	0.616	0.048	0.381–0.996
	pTRF	−0.3	0.741	0.044	0.553–0.992					
	pEI	0.047	1.048	0.118	0.988–1.111					
	pSRE	0.719	2.052	0.192	0.697–6.044	Age	−0.14	0.696	0.92	0.456–1.060
	pEI to TRF ratio	0.65	1.916	0.071	0.947–3.877	Male	0.262	1.300	0.807	0.158–10.682
	npThickness of Fat Layer	0.067	1.069	0.45	0.899–1.271	Body Mass Index	−0.62	0.940	0.772	0.670–1.320
	npTRF	−0.357	0.699	0.021	0.516–0.948	pTRF	−0.363	0.696	0.92	0.456–1.060
	npEI	0.027	1.028	0.289	0.977–1.081					
	npSRE	0.783	2.189	0.203	0.656–7.307					
	npEI to TRF ratio	0.466	1.593	0.134	0.866–2.932					

p, paretic; np, non-paretic; EI, echo intensity; TRF, thickness of rectus femoris; SRE, strain ratio of elastography.

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
