# Peer review of "Ultrasound Evaluation of the Rectus Femoris for Sarcopenia in Patients with Early Subacute Stroke"

_jcm, 2021, doi:10.3390/jcm10143010_

Round 1
Reviewer 1 Report
The manuscript by Choi et al. is very interesting since it investigates the possibility to evaluate the presence of alterations detactable with US at the bedside of the patient to infer about the presence of low ALM. The issue is of great value since it would be very useful in everyday clinical practice to have an easy-to-obtain indicator or predictor of loss of ALM.
The aim of the study was to study the rectus femoris with US in subacute stroke patients to evaluate if it could assist in detecting factors affecting appendicular lean muscle loss, and to elucidate the relationship between ultrasound and dual-energy X-ray absorptiometry measurement for ALM in the early subacute stroke phase.
The article is detailed and well written, but it is not easy to read. In particular, I found that the results seems to be redundant and the conclusions are not congruent with the results.
Main criticism:
- methodological issue: it is not possible to investigate “early muscle changes” or “ALM loss” if only one measure is performed. Indeed, to assess early muscle changes or ALM loss at least 2 observations performed distantly in time (ideally one at baseline in the hyperacute phase) are expected. Besides, in the title the Authors talk about “Early Subacute Stroke-Related Low Appendicular Lean Mass”. I think that this study does not allow to talk about nor to an early subacute or a stroke-related effect.
Other criticisms:
- the Authors analysed the ALM both as a dichotomous variable and as a continuous one, thus performing respectively t-test/linear regression analyses, and correlation/logistic regression analyses. I think that this only generates confusion. Besides, if some results are significant only when treating ALM as continuous and not as dichotomous variable, maybe I would assume that the association is not very strong and clinically relevant.
- in the conclusion the Authors report that EI/TRF on the hemiplegic side was a key factor for early detection of myosteatosis and evaluation of ALM in the early subacute stroke phase. However, the only independently associated variable in regression analyses was TRF on non-hemiplegic side.
- the Authors did not report about ischemic stroke details. Which was the NIHSS at arrival in emergency room and at the moment of US measurement? What was the mRS scale at the moment of US measurement? How many patients in each group had weakness in the leg and what was the corresponding MRC scale? Besides, we ususally define hemiplegic a side of the body where no movement are appreciated: was this the case or the Authors are talking about different degrees of weakness in the affected side of the body?
Author Response
Main criticism:
- methodological issue: it is not possible to investigate “early muscle changes” or “ALM loss” if only one measure is performed. Indeed, to assess early muscle changes or ALM loss at least 2 observations performed distantly in time (ideally one at baseline in the hyperacute phase) are expected. Besides, in the title the Authors talk about “Early Subacute Stroke-Related Low Appendicular Lean Mass”. I think that this study does not allow to talk about nor to an early subacute or a stroke-related effect.
(Greeting)
Firstly, we appreciate your criticisms. We can greatly improve our manuscript based on the issues raised.
(Answer)
This study is about ultrasound characteristics of the rectus femoris in patients with early subacute stroke with identified sarcopenia.
Patients with early subacute stroke are first met after neurosurgery or acute treatment in neurosurgery or neurology part.
However, there are currently few reports on the ultrasound characteristics of the femoral muscle.
As the Reviewer stated, at least two measurements would be ideal.
However, the authors felt that the cross-sectional design in this study is realistic considering the clinical approach.
Therefore, in this study, we tried to minimize variables such as those mentioned by the Reviewer with strict inclusion and exclusion criteria.
- Inclusion criteria:
- Patients with subacute stroke for the first time whose lesions were confirmed using brain imaging (MRI/CT)
- Patients who had unilateral hemiplegia
- Patients who can follow 1 step of the Mini-Mental State Examination in Korean K ≥ 10
- Patients who did not have difficulty in premorbid ambulation or activities of daily living without devices
- Patients who can eat orally
- Patients with Functional Ambulation Category scores of at least 2.
- Exclusion criteria:
- Patients who do not meet the inclusion criteria
- Patients with double hemiplegia
- Patients with amputated lower limbs
- Patients with a fracture or surgical history in lower limbs that led to gait disturbance
- Patients with any surgeries that led to malnutrition
- Patients with other systemic diseases that can affect the loss of muscle mass or malnutrition (hypogonadism, hypercortisolism, hyperthyroidism, growth hormone deficiency, chronic obstruction pulmonary disease, chronic kidney disease, liver cirrhosis, chronic heart failure, neurodegenerative diseases, inflammatory diseases, acquired immune deficiency syndrome, and malignancy)
We have modified the criteria to be more explicit.
As per the Reviewer’s remark, it cannot be ruled out that the patient with sarcopenia in this study developed it before stroke, thus it was mentioned as a limitation. We will try to supplement it in future research.
And we change the title to Ultrasound evaluation of the rectus femoris for sarcopenia in patients with early subacute stroke.
Other criticisms:
- the Authors analysed the ALM both as a dichotomous variable and as a continuous one, thus performing respectively t-test/linear regression analyses, and correlation/logistic regression analyses. I think that this only generates confusion. Besides, if some results are significant only when treating ALM as continuous and not as dichotomous variable, maybe I would assume that the association is not very strong and clinically relevant.
(Answer)
In agreement with the Reviewer's opinion, the literature was reassessed to include content related to dichotomous variables.
The Reviewer's advice has made this article more concise and clearer.
- in the conclusion the Authors report that EI/TRF on the hemiplegic side was a key factor for early detection of myosteatosis and evaluation of ALM in the early subacute stroke phase. However, the only independently associated variable in regression analyses was TRF on non-hemiplegic side.
(Answer)
We agree with the Reviewer’s criticism.
There was an error when writing the abstract.
We have revised the relevant contents.
- the Authors did not report about ischemic stroke details. Which was the NIHSS at arrival in emergency room and at the moment of US measurement? What was the mRS scale at the moment of US measurement? How many patients in each group had weakness in the leg and what was the corresponding MRC scale? Besides, we ususally define hemiplegic a side of the body where no movement are appreciated: was this the case or the Authors are talking about different degrees of weakness in the affected side of the body?
(Answer)
We included first-ever cases of subacute stroke with unilateral hemiplegia in patients aged older than 40 years.
All measurements were evaluated at the time of transfer. The evaluation time was recorded as post onset. FAC was used in this study. Both mRS and FAC are assessments of ambulation ability.
The mRS of the participating patients was between 1 and 4, and the FAC between 2 and 5.
According to the Reviewer's opinion, hemiplegia was changed to paresis or paretic side.
Thank you for taking the time out of your busy schedule to review our manuscript.
I hope that you find yourself in good health during Covid-19.

Reviewer 2 Report
Comments to the author:
Interesting topic - thank you for your efforts in contributing to the literature on this topic. However, there are methodological weaknesses. Many results are presented. However, the red thread is missing due to the quantity. The results should be presented in a more focused way. Subordinate results could be moved to the Supplemental Material section.
Nevertheless, a good idea for early evaluation of patients with stroke.
I have provided a few thoughts for your consideration or modification.
General comments:
A lot of abbreviations were used. The manuscript is difficult read and understand. Please, minimize the abbreviations to a minimum.
The inclusion and grouping of study participants appears unclear. Which groups or which aspects were compared are incomplete or vaguely formulated. This should be clarified in the method section.
Detailed comments:
Abstract:
(3) Results: Do you mean “ALM” or “ALM-loss”? The abbreviations should be clarified in in the abstract and entire manuscript.
Please, add main results with p-values.
Introduction
Background related to elastography (muscle stiffness) might be introduced.
The Hypothesis section remains unclear. Hypothesis are needed instead of mentioning the potential of a method. The hypothesis related to changes in stiffness remain unclear.
The aim of the study should be revised. Why is the comparison with DAXA useful?
Methods
Subjects: Groups remain unclear. What exactly was compared? I guess low-ALM and intact-ALM. Or hemiplegic vs non- hemiplegic? Group definitions?
“Sonographer was blinded to clinical presentation”. This appears confusing. Maybe, the sonographer was blinded to the final diagnosis?
Please, add the software versions.
Elastography: How were optimal images defined? How many measurement were performed and averaged? The experience level in elastography (and US) of the examiner might be mentioned. Were the elastography measurements performed in longitudinal or transversal plane?
Results
A more focused demonstration of the results based on the hypothesis might appreciated.
Figures of the study group as well as control might be added.
Discussion
A comparison of the absolute values should be compared with those in the literature.
Elastography is not elaborated further. What is your explanation for your finding?
A summary of the study strength is not essentially needed.
Author Response
Interesting topic - thank you for your efforts in contributing to the literature on this topic. However, there are methodological weaknesses. Many results are presented. However, the red thread is missing due to the quantity. The results should be presented in a more focused way. Subordinate results could be moved to the Supplemental Material section.
Nevertheless, a good idea for early evaluation of patients with stroke.
I have provided a few thoughts for your consideration or modification.
General comments:
A lot of abbreviations were used. The manuscript is difficult read and understand. Please, minimize the abbreviations to a minimum.
(Greeting)
Firstly, we appreciate your criticisms. We were able to write a more focused manuscript based on your opinions.
(Answer)
Based on the comments of the Reviewer, the use of abbreviations was modified to a minimum.
The inclusion and grouping of study participants appears unclear. Which groups or which aspects were compared are incomplete or vaguely formulated. This should be clarified in the method section.
(Answer)
Inclusion and exclusion criteria are now clearly stated based on the Reviewers' comments.
Low ALM was changed to sarcopenia.
It was rewritten as a comparison between the sarcopenia and normal groups.
Detailed comments:
Abstract:
(3) Results: Do you mean “ALM” or “ALM-loss”? The abbreviations should be clarified in in the abstract and entire manuscript.
Please, add main results with p-values.
(Answer)
Low ALM was replaced with sarcopenia.
We tried to minimize the use of abbreviations.
The main results were rewritten to include p-values in the abstract and manuscript.
Introduction
Background related to elastography (muscle stiffness) might be introduced.
The Hypothesis section remains unclear. Hypothesis are needed instead of mentioning the potential of a method. The hypothesis related to changes in stiffness remain unclear.
The aim of the study should be revised. Why is the comparison with DAXA useful?
(Answer)
This study is about ultrasound characteristics of the rectus femoris in patients with early subacute stroke with sarcopenia identified by DEXA.
DEXA is the golden standard test for sarcopenia.
Echo intensity, echo intensity/thickness of the rectus femoris, and strain elastography ratio were evaluated to confirm the qualitative degenerative change of the muscle.
Degenerated muscles increased echo intensity depending on intramuscular fat infiltration. The thickness of the rectus femoris was reduced depending on the loss of lean mass in the lower limbs.
It was amended as follows.
“We assumed that the lower the ALM, the worse the degeneration in the rectus femoris would become. An ultrasound study would be able to detect degenerative changes in the rectus femoris in the early subacute stroke period. TRF would be decreased according to the loss of ALM. EI would be increased according to fatty infiltration into the muscle. Therefore, EI to TRF ratio would be clearly increased according to the severity of myosteatosis. SRE would change with muscle degeneration. But there were few reports on these in the early subacute stroke period. We thus evaluated the ultrasound characteristic of the rectus femoris for sarcopenia measured by DEXA and compared it patients without sarcopenia in the early subacute stroke phase.”
Methods
Subjects: Groups remain unclear. What exactly was compared? I guess low-ALM and intact-ALM. Or hemiplegic vs non- hemiplegic? Group definitions?
(Answer)
The meaning was not clearly conveyed.
Low ALM was changed to the sarcopenia group.
Intact-ALM was changed to the normal group.
It was rewritten as a comparison between the sarcopenia and normal groups.
“Sonographer was blinded to clinical presentation”. This appears confusing. Maybe, the sonographer was blinded to the final diagnosis?
(Answer)
It meant that the "sonographer was not aware of patient information."
This sentence was modified to make the meaning clear.
Please, add the software versions.
(Answer)
The software version was added according to the Reviewer's comment.
Elastography: How were optimal images defined? How many measurement were performed and averaged? The experience level in elastography (and US) of the examiner might be mentioned. Were the elastography measurements performed in longitudinal or transversal plane?
(Answer)
The sonographer had three years of experience in musculoskeletal ultrasound.
Strain elastography obtained a constant color scale image and compression bar.
The image of elastography was auto-selected by a US machine.
Three consecutive static images were acquired and averaged to minimize measurement variability
Results
A more focused demonstration of the results based on the hypothesis might appreciated.
Figures of the study group as well as control might be added.
(Answer)
Revisions were made based on the Reviewer’s comments.
Ultrasound images of the sarcopenia and normal groups were added.
Discussion
A comparison of the absolute values should be compared with those in the literature.
Elastography is not elaborated further. What is your explanation for your finding?
A summary of the study strength is not essentially needed.
(Answer)
In this study, TRF showed a significant change between the two groups, but EI showed no significant difference between the two groups, only tending to increase with degenerative changes in the muscle.
Changes in EI and TRF moved in opposite directions according to the degenerative changes in the muscle.
Therefore, even when intramuscular fat deposition was not clearly observed, the EI to TRF ratio was sensitively reflected according to the degenerative changes in the muscle in the early subacute stroke phase.
There was no significant difference in SRE between the two groups.
The strain ratio of elastography is a value comparing the elasticity of subcutaneous fat with that of the rectus femoris muscle.
There was no significant difference between the two groups in the amount of fat mass assessed by DEXA and the thickness of the fat layer measured by ultrasound.
Intramuscular fat infiltration increased the stiffness of the muscle. However, considering that only the increasing tendency of EI was confirmed, this was the starting point for intramuscular fat deposition in this study.
We thought that SRE did not reflect sensitive changes, unlike the EI to TRF ratio, because there was little change in the fat layer, and because muscle stiffness was not sufficiently increased depending on intramuscular fat infiltration in this study.
In addition to muscle degenerative changes, the strain ratio of elastography increased according to spasticity. There was no effect on spasticity because the modified Ashworth scale of the patients in this study was zero.
The above information was added in the discussion.
The summary of the study was deleted according to the Reviewer's opinion.
Thank you for taking the time out of your busy schedule to review our manuscript.
I hope that you find yourself in good health during Covid-19.

Reviewer 3 Report
In this manuscript, the authors examined whether observation of the rectus femoris by ultrasound could assist in detecting factors affecting appendicular lean muscle loss. This manuscript is generally well written. The methods are adequately described and reasonable. The conclusion is well supported by the results. I have only some minor concerns.
- Has the hand grip test been performed? The hand grip test is one of important tests to evaluate sarcopenia. (The Journals of Gerontology: Series A, Volume 69, Issue 5, May 2014, Pages 576–583). I think that the hand grip test was done along with other tests such as MBI or FAC. I would like to know about this.
- There are some typo errors.
Table 4: TF -> FT
Table 5: nhS -> nhSRE
Author Response
In this manuscript, the authors examined whether observation of the rectus femoris by ultrasound could assist in detecting factors affecting appendicular lean muscle loss. This manuscript is generally well written. The methods are adequately described and reasonable. The conclusion is well supported by the results. I have only some minor concerns.
- Has the hand grip test been performed? The hand grip test is one of important tests to evaluate sarcopenia. (The Journals of Gerontology: Series A, Volume 69, Issue 5, May 2014, Pages 576–583). I think that the hand grip test was done along with other tests such as MBI or FAC. I would like to know about this.
(Greeting)
Firstly, we appreciate your criticisms. We wrote a more complete article based on your opinions.
(Answer)
As the Reviewer assumed, all patients who participated in this study performed a grip test.
The result of the grip test depends on the dominant hand.
Left-handed individuals do not differ in left and right grip strength, but right-handed individuals have a higher grip strength than do left-handed individuals. (Petersen, Paul, et al. "Grip strength and hand dominance: challenging the 10% rule." American Journal of Occupational Therapy 43.7 (1989): 444-447.)
The dominant hand of the patients involved in this study was the right hand. The patient developed left or right side paralysis.
Measurements performed with the left or right hand could not be assessed equally.
Therefore, grip tests were excluded from this study.
- There were some typographical errors.
Table 4: TF -> FT
Table 5: nhS -> nhSRE
(Answer)
The tables were deleted in response to the other Reviewers' comments.
Thank you for taking the time out of your busy schedule to review our manuscript.
I hope that you find yourself in good health during Covid-19.

Round 2
Reviewer 2 Report
Thank you for the intensive time you took to edit the manuscript. My comments were all implemented.
I have provided minor thoughts for your consideration or modification.
Minor comments:
Methods:
“The exclusion criteria were: patients who did not meet the inclusion criteria; …” – not meeting the inclusion criteria should not be part of the exclusion criteria and can be removed at this point.
Please, add a reference for automated elastography measurement.
Figure 1 should be mentioned with the results.
Figure 1: the target structures (muscle) should be marked.
Author Response
Dear Reviewer
Thank you for your detailed review for a better manuscript.
The manuscript has been rechecked and the necessary changes have been made in accordance with the reviewers’ suggestions. The responses to all comments were written below.
Minor comments:
Methods:
“The exclusion criteria were: patients who did not meet the inclusion criteria; …” – not meeting the inclusion criteria should not be part of the exclusion criteria and can be removed at this point.
(Answer)
In agreement with the Reviewer's opinion, the sentence was removed.
Please, add a reference for automated elastography measurement.
(Answer)
We added the reference. [27]
Yamamoto, Y.; Yamaguchi, S.; Sasho, T.; Fukawa, T.; Akatsu, Y.; Nagashima, K.; Takahashi, K. Quantitative Ultrasound Elastography With an Acoustic Coupler for Achilles Tendon Elasticity: Measurement Repeatability and Normative Values. Journal of ultrasound in medicine : official journal of the American Institute of Ultrasound in Medicine 2016, 35, 159-166, doi:10.7863/ultra.14.11042.
Figure 1 should be mentioned with the results.
Figure 1: the target structures (muscle) should be marked.
(Answer)
Based on the comments of the Reviewer, the relevant contents were revised.
(Add on)
We changed to non-sarcopenia group rather than normal group instead of intact-ALM group.
